# Role of Compensatory miRNA Networks in Cognitive Recovery from Heart Failure

**DOI:** 10.3390/ncrna11030045

**Published:** 2025-06-12

**Authors:** Verena Gisa, Md Rezaul Islam, Dawid Lbik, Raoul Maximilian Hofmann, Tonatiuh Pena, Dennis Manfred Krüger, Susanne Burkhardt, Anna-Lena Schütz, Farahnaz Sananbenesi, Karl Toischer, Andre Fischer

**Affiliations:** 1Department for Epigenetics and Systems Medicine in Neurodegenerative Diseases, German Center for Neurodegenerative Diseases, Von Siebold Street 3A, 37075 Goettingen, Germany; 2Clinic of Cardiology and Pneumology, Georg-August-University, Robert-Koch Street 38, 37075 Goettingen, Germany; 3Research Group for Genome Dynamics in Brain Diseases, German Center for Neurodegenerative Diseases, Von Siebold Street 3A, 37075 Göttingen, Germany; 4Cluster of Excellence “Multiscale Bioimaging: from Molecular Machines to Networks of Excitable Cells” (MBExC), Von Siebodl Street 3A, 37077 Göttingen, Germany; 5DZKH (German Center for Cardiovascular Diseases), Robert Koch Street 40, 37075 Göttingen, Germany; 6Clinic of Psychiatry and Psychotherapy, University Medical Center, Von Siebold Street 5, 37075 Goettingen, Germany

**Keywords:** heart failure, cognitive impairment, hippocampal function, MicroRNA, transcriptional homeostasis, memory recovery, Alzheimer

## Abstract

**Background:** Heart failure (HF) is associated with an increased risk of cognitive impairment and hippocampal dysfunction, yet the underlying molecular mechanisms remain poorly understood. This study aims to investigate the role of microRNA (miRNA) networks in hippocampus-dependent memory recovery in a mouse model of HF. **Methods:** CaMKIIδC transgenic (TG) mice, a model for HF, were used to assess hippocampal function at 3 and 6 months of age. Memory performance was evaluated using hippocampus-dependent behavioral tasks. Small RNA sequencing was performed to analyze hippocampal miRNA expression profiles across both time points. Bioinformatic analyses identified miRNAs that potentially regulate genes previously implicated in HF-induced cognitive impairment. **Results:** We have previously shown that at 3 months of age, CaMKIIδC TG mice exhibited significant memory deficits associated with dysregulated hippocampal gene expression. In this study, we showed that these impairments, memory impairment and hippocampal gene expression, were no longer detectable at 6 months, despite persistent cardiac dysfunction. However, small RNA sequencing revealed a dynamic shift in hippocampal miRNA expression, identifying 27 miRNAs as “compensatory miRs” that targeted 73% of the transcripts dysregulated at 3 months but reinstated by 6 months. Notably, miR-181a-5p emerged as a central regulatory hub, with its downregulation coinciding with restored memory function. **Conclusions:** These findings suggest that miRNA networks contribute to the restoration of hippocampal function in HF despite continued cardiac pathology and provide an important compensatory mechanism towards memory impairment. A better understanding of these compensatory miRNA mechanisms may provide novel therapeutic targets for managing HF-related cognitive dysfunction.

## 1. Introduction

Heart failure (HF) is a leading cause of morbidity and mortality worldwide, affecting millions of people across all age groups [1]. It is a chronic and progressive condition characterized by the inability of the heart to pump blood effectively, leading to systemic effects that extend far beyond the cardiovascular system. Importantly, HF has emerged as a significant risk factor for cognitive decline and neurodegenerative diseases, including Alzheimer’s disease (AD) [2].

Patients with HF are at increased risk of developing hippocampal dysfunction, resulting in deficits in learning and memory processes [3]. However, the molecular mechanisms underlying this relationship remain poorly understood. Several studies reported that heart failure leads to altered gene expression patterns in the brain, and other organs [4,5,6]. In a previous study, we established a strong correlation between cardiac dysfunction and cognitive impairment in a mouse model of HF driven by CaMKIIδC overexpression [6]. Specifically, 3-month-old CaMKIIδC transgenic (TG) mice exhibited significant impairments in hippocampus-dependent memory formation, accompanied by extensive transcriptional dysregulation in hippocampal tissues.

In this study, we analyzed the CaMKIIδC mouse model for HF at 6 months of age and made a surprising observation: memory deficits were no longer apparent, despite the persistence of severe cardiac dysfunction. This raises the intriguing possibility of a compensatory mechanism that restores hippocampal function and homeostasis. At the molecular level, microRNAs (miRNAs) have been established as key regulators of transcriptional and post-transcriptional homeostasis. miRNAs are small, non-coding RNAs that act as fine-tuners of gene expression by binding to complementary mRNA sequences, leading to mRNA degradation or translational repression [7,8]. By buffering fluctuations in gene expression, miRNAs ensure cellular stability and play critical roles in neuronal plasticity, memory formation, and stress responses [9,10]. Given their ability to coordinate the expression of multiple target genes, miRNAs are well-suited to mediate compensatory mechanisms in disease conditions.

Here, we identify a compensatory hippocampal miRNA signature that restores transcriptional homeostasis and may contribute to the rescue of memory deficits in HF. These findings provide new insights into the molecular mechanisms underlying brain resilience and suggest that targeting specific miRNAs may represent a novel therapeutic approach to treating cognitive decline in HF patients.

## 2. Results

### 2.1. Six-Month-Old CamKIIδC Mice Exhibit No Memory Impairments Despite Heart Failure

We first confirmed that 6-month-old CamKIIδC mice exhibit significant cardiac dysfunction, as previously reported. Compared to control mice, CamKIIδC mice showed reduced ejection fraction (Figure 1A), cardiac output (Figure 1B), and cardiac index (Figure 1C). Additionally, the heart/body weight ratio (Figure 1D), left ventricle (LV)/body weight ratio (Figure 1E), and lung/body weight ratio (Figure 1F) were significantly increased, indicating heart failure. Body weight remained unchanged between groups (Figure 1G). These data are in agreement with previous studies [11].

Despite this cardiac dysfunction, hippocampus-dependent learning and memory were not impaired in 6-month-old CamKIIδC mice, as assessed using the Barnes maze test. Escape latency decreased similarly in CamKIIδC and control mice over 7 training days (Figure 2A). On day 8, both groups spent comparable time at the target hole (Figure 2B) and within the target quadrant (Figure 2C,D), indicating intact memory.

To exclude the confounding effects of anxiety or exploratory behavior, we performed an open field test. No significant differences were observed in total distance traveled (Figure 2E), travel speed (Figure 2F), or time spent in the center area (Figure 2G,H), confirming normal exploratory behavior and baseline anxiety.

These results were unexpected, as 3-month-old CamKIIδC mice showed both cardiac dysfunction and impaired memory function [6]. This suggests that a compensatory mechanism may restore hippocampal function in 6-month-old mice.

### 2.2. Hippocampal Gene Expression Reveals a Potential Compensatory Mechanism

We previously reported that memory impairment in 3-month-old CamKIIδC mice was correlated with significant changes in hippocampal gene expression [6]. To better understand the absence of memory impairment in 6-month-old CamKIIδC mice, we compared hippocampal gene expression in CamKIIδC and control mice at both 3 and 6 months of age. Small and total RNA sequencing (RNA-seq) datasets had been generated previously [6], but the data from the 6-month-old mice had not yet been analyzed. Therefore, we re-analyzed all datasets together.

We compared hippocampal gene expression between 3- and 6-month-old CamKIIδC TG mice and controls. At 3 months, we identified 689 differentially expressed genes (DEGs; 122 up-regulated and 567 down-regulated) in CamKIIδC TG mice (FDR < 0.05, log2FC > 0.1) when compared to the control group. Using a stricter cutoff (log2FC > 0.26), 246 downregulated and 52 upregulated genes were detected (Figure 3A; Appendix A). As described before, Gene-Ontology (GO)-term analysis revealed that the down-regulated genes were enriched for processes linked to memory function, such as “synaptic transmission” and pathways such as “long-term potentiation” (Appendix A). In contrast, only 11 DEGs were detected at 6 months (FDR < 0.05, log2FC > 0.1), with just 4 genes meeting the stricter cutoff (Figure 3A). This indicates that hippocampal transcriptional dysregulation observed at 3 months is largely resolved by 6 months (Figure 3B and Appendix A), potentially explaining the restored memory function.

To better understand the potential compensatory gene-expression response occurring between 3 and 6 months of age, we directly compared these two time points using differential expression analysis in both wild-type control and CamKIIδC mice. In wild-type control mice, we identified 223 differentially expressed genes between 3 and 6 months of age (Figure 3C,D; Appendix A). In contrast, the same comparison in CamKIIδC TG mice revealed 935 differentially expressed genes (Figure 3C,E; Appendix A).

These results indicate that only minor changes in hippocampal gene expression occur in wild-type control mice over this time period, whereas CamKIIδC TG mice exhibit substantial transcriptional changes. This supports the hypothesis that a compensatory mechanism in CamKIIδC TG mice drives differential gene expression during aging, ultimately reinstating physiological gene-expression levels. As expected, no such process is observed in wild-type control mice.

### 2.3. A Compensatory miRNA Network May Regulate Gene Expression Recovery

MicroRNAs are known to regulate gene expression at the systems level, with one miRNA being able to control multiple transcripts within a signaling pathway [12]. Thus, miRNAs are recognized as important regulators of cellular homeostasis [13]. By fine-tuning mRNA and protein levels, miRNAs can buffer against fluctuations in gene expression, ensuring stability within cellular networks. Therefore, we hypothesized that altered miRNA expression could mediate the observed compensatory gene-expression response, at least in part.

To test this hypothesis, we subjected RNA isolated from the hippocampus of 3- and 6-month-old wild-type control and CamKIIδC TG mice to small RNA sequencing. We observed only minor changes in miRNA expression in 3-month-old mice comparing wild-type control mice and CamKIIδC TG mice, and no changes in miRNA expression in 6 6-month-old mice (Appendix A). Differential expression analysis revealed that 26 miRNAs were significantly altered (FDR < 0.05, log2FC > 0.26) in wild-type control mice, with 5 miRNAs downregulated and 21 miRNAs upregulated when comparing the tissue from mice at 3 and 6 months of age (Figure 4A; Appendix A). In contrast, analysis of CamKIIδC mice revealed 221 miRNAs with significantly altered expression profiles between 3 and 6 months, of which 124 miRNAs were upregulated and 97 miRNAs were downregulated (Figure 4B; Appendix A). These results indicate that hippocampal miRNA levels exhibit only minor changes during this time period in wild-type control mice. However, in CamKIIδC mice, substantial changes in miRNA expression occur between 3 and 6 months of age.

Next, we tested the hypothesis that miRNAs with altered expression between 3- and 6-month-old CamKIIδC TG mice may target mRNA transcripts that were downregulated in 3-month-old CamKIIδC mice relative to wild-type controls. Since most of the deregulated mRNA transcripts in 3-month-old CamKIIδC TG mice were decreased compared to controls, we focused on this subset of genes. Specifically, we aimed to identify miRNAs that were significantly downregulated between 3 and 6 months in CamKIIδC mice and determine how many of these miRNAs target the downregulated mRNA transcripts that exhibit reinstated expression levels at 6 months (Figure 5A). For this approach, we defined deregulated genes as reinstated if their expression increased by more than 50%, corresponding to a difference in the normalized fold change greater than 1.5 (Appendix A). This criterion was met by 43% of the deregulated genes. Next, we examined whether any miRNAs significantly downregulated between 3- and 6-month-old CamKIIδC mice target these reinstated transcripts (Appendix A). Notably, 73% (*n* = 94) of the reinstated genes were targets of these compensatory miRNAs, a group we referred to as “rescued transcripts” (Figure 5B; Appendix A). Further analysis revealed that the rescued transcripts were mainly regulated by 27 miRNAs, which we termed “compensatory miRs” (Figure 5C). Among the miRNAs with the most targets was miR-181a-5p, a miRNA previously implicated in neuronal plasticity and memory formation [14] (Figure 5C). Other members of the miR-181 family, including miR-181b-5p and miR-181d-5p, were also part of the compensatory miRs. Additionally, the majority of miRNAs from the let-7 family (let-7c-5p, let-7i-5p, let-7a-5p, let-7g-5p, let-7e-5p, let-7d-5p, and miR-98-5p) were identified among the 27 compensatory miRs. Similarly, two members of the miR-29 family (miR-29b-3p and miR-29c-3p), both members of the miR-92 family (miR-92a-3p and miR-92b-3p), and three members of the miR-23/27/24 cluster (miR-27b-3p, miR-23a-3p, and miR-23b-3p) were included. Other notable compensatory miRNAs included miR-1a-3p, miR-667-5p, miR-136-5p, miR-125b-5p, and miR-153-3p (Figure 5C). Next, we performed a GO term analysis of the reinstated genes that were targets of the compensatory miRs. Interestingly, genes that were upregulated from 3 to 6 months in CamKIIδC mice were linked to processes related to RNA metabolism and mRNA expression (Figure 5D; Appendix A), which aligns with the notion that the hippocampus engages the compensatory miR network to reinstate physiological gene expression. To gain deeper insights into the functional role of these compensatory miRNAs, we also constructed a transcriptional interaction network based on the 27 miRNAs and 77 rescued genes upregulated in 6-month-old CamKIIδC mice compared to 3-month-old (Figure 5E). This network revealed that the 27 compensatory miRNAs synergistically regulate the expression of rescued transcripts and identified miR-181a-5p as a hub miRNA orchestrating the compensatory response.

## 3. Discussion

In this study, we investigated the molecular mechanisms linking HF to cognitive impairment. Using CaMKIIδC TG mice, we previously demonstrated significant memory impairment at 3 months of age, accompanied by extensive dysregulation of hippocampal gene expression [6]. These data are consistent with epidemiological studies showing that cardiac disease increases the risk of age-associated memory impairment [15,16,17]. Surprisingly, by 6 months of age, memory deficits were no longer detectable in CaMKIIδC TG mice, suggesting the activation of compensatory mechanisms that restore hippocampal function and transcriptional homeostasis. Therefore, we studied the underlying mechanisms that could help to explain the recovery of hippocampus-dependent memory function in this mouse model of HF, despite the persistence of cardiac dysfunction.

We observed that while hippocampal gene expression was deregulated in 3-month-old CaMKIIδC TG mice, no such differences were detected at 6 months of age. However, molecular changes were not entirely absent in the hippocampus of 6-month-old CaMKIIδC TG mice, as miRNAs exhibited altered expression levels at this time point, suggesting a potential role for a compensatory miRNA network in driving recovery. Specifically, small RNA sequencing revealed significant changes in miRNA expression between 3 and 6 months in CaMKIIδC TG mice, whereas only minor changes were observed in age-matched wild-type controls.

By focusing on mRNA transcripts downregulated at 3 months—which represent the majority of deregulated genes—we found that 43% of these genes exhibited near-complete reinstatement of expression levels at 6 months, suggesting partial recovery of transcriptional activity. Notably, 27 miRNAs, which we termed “compensatory miRs”, were downregulated between 3 and 6 months and targeted 73% of these reinstated transcripts, indicating a critical role for these miRNAs in rescuing gene expression.

Compensatory miRNA responses have been described in various disease contexts, such as responses to cellular stressors like hypoxia and oxidative damage [18,19,20,21,22]. Our current findings extend this concept to HF-induced memory impairment. Among the identified 27 miRNAs, miR-181a-5p emerged as a central hub miRNA within the compensatory network. miR-181a-5p was decreased in the hippocampus of 6-month-old CaMKIIδC TG mice, consistent with previous findings showing that elevated levels of miR-181a-5p are associated with impaired neuronal plasticity, synaptic function, and memory formation [14,23,24]. In turn, inhibition of miR-181a-5p was shown to reinstate memory formation in a mouse model of AD, and its downregulation was linked to improved neuronal integrity in a calorie restriction model that ameliorated age-associated memory impairment in mice [25]. Furthermore, elevated levels of miR-181a-5p were implicated in cell death after cerebral ischemia, while reduced levels were associated with neuronal survival [26,27,28], supporting the notion that lowering miR-181a-5p levels in disease contexts can improve memory function. These data strongly suggest that its downregulation in CaMKIIδC TG mice between 3 and 6 months likely contributes to the reinstatement of learning ability. Interestingly, additional members of the miR-181 family, including miR-181b-5p and miR-181d-5p, were also part of this network, suggesting a coordinated role for the miR-181 family in mediating the compensatory response.

Notably, other miRs of the identified “compensatory miR” network represented almost entire miRNA families. For example, we identified several other miRNA families, including the let-7 family and the miR-29 family, which targeted the reinstated transcripts. These miRNAs have been associated with neuroprotection and synaptic remodeling, further supporting their role in restoring hippocampal function [29,30]. Moreover, members of the miR-92 [31], miR-29 [32,33], and miR23/27/24 clusters [34,35,36], which were part of the “compensatory miR” network, have documented roles in neuronal function and learning behavior. The same is true for miR-1a-3p [37], miR-136-5p [38], miR-125b-5p [39,40], and miR-153-3p [41]. Collectively, these findings highlight the synergistic action of multiple miRNAs in regulating hippocampal gene expression to compensate for early transcriptional deficits.

miRNAs are also discussed as potential biomarkers for cognitive function and neurodegenerative diseases [14,42], though data in this area are sometimes conflicting. For instance, some studies reported that certain miRNAs (e.g., miR-146a) are upregulated in AD patients [14,24], while others found them to be downregulated [43]. If such miRNAs are part of a compensatory response, as indicated by our data, longitudinal measurements of miRNA expression will be crucial for resolving these discrepancies.

Our study has several limitations that warrant consideration and should be addressed in future research. First, while our findings identify a compensatory miRNA network that potentially mediates the restoration of hippocampal function in HF, the specific mechanisms regulating these miRNAs remain unclear. The observation that most compensatory miRs belong to miRNA families acting on similar pathways suggests a concerted action, tightly linking mRNA transcript levels to miRNA regulation. Such mechanisms are well-documented in developmental processes, where miRNAs regulate mRNA levels through feed-forward and feedback loops to stabilize developmental pathways [44]. Nevertheless, our data only provides a model and future research should explore how specific transcription factors or epigenetic processes, such as DNA methylation or histone modifications, contribute to the regulation of this compensatory miRNA network.

While this study focused on hippocampal miRNA and gene expression, HF is a systemic condition likely affecting other brain regions critical for cognitive function. Future research should investigate whether similar compensatory mechanisms operate in other brain areas. Moreover, although we used a well-characterized mouse model of HF, validating these findings in additional models of HF will be important for generalizability. Furthermore, while this study primarily explored transcriptional and miRNA landscapes, other regulatory mechanisms, such as protein–protein interactions and metabolomic changes, may also contribute to the observed recovery of memory function. Employing multi-omics approaches could provide a more comprehensive understanding of the compensatory response. It would also be important to compare the mechanisms underlying the recovery of memory function in this study to those involved in other interventions. For example, aerobic exercise can improve cognitive function in heart failure patients, especially in those who already show cognitive impairment [45]. At the same time, there is evidence that the beneficial effect of exercise on cognitive function is—at least in part—mediated via adaptive changes in miRNA expression [46,47].

Finally, the long-term sustainability of the observed compensatory mechanisms remains uncertain. While hippocampal function appears restored at 6 months, compensatory mechanisms may eventually fail, leading to late-onset cognitive decline. The unexpected observation that 3-month-old CaMKIIδC TG mice displayed memory impairment, which was no longer evident at 6 months, underscores the need for longitudinal studies. Examining CaMKIIδC TG mice at later time points and expanding such analyses to other models of age-associated cognitive decline will be essential.

In conclusion, while our findings provide significant insights into the role of miRNA networks in restoring cognitive function in the context of HF, addressing these remaining questions through future research will refine our understanding and help identify novel therapeutic targets for managing heart failure-associated cognitive dysfunction.

## 4. Materials and Methods

### 4.1. Animals and Tissue Preparation

For animal experiments, mice with a genetic background of C57Bl/6J were used. CamkIIδc transgenic and wild-type littermates were housed in standard cages on a 12 h/12 h light/dark cycle with food and water ad libitum. Three- and Six-month-old mice were analyzed for the experiments. Male and female mice were used in the experiments. Following cervical dislocation, the hippocampal sub-region CA1 was isolated. Hearts were dissected by a cut above the base of the aorta and perfused with 0.9% sodium chloride solution until blood-free. Tissues were snap-frozen in liquid nitrogen after collection and stored at −80 °C. In addition, the lung was extracted, and its weight was determined.

### 4.2. Echocardiography

Heart function and dimensions were assessed via echocardiography using a Vevo 2100 imaging platform equipped with a MS-400 30 MHz transducer(Visualsonics, Toronto, ON, Canada). The animals were anesthetized with isoflurane (1–2%), and M-mode sequences of the beating heart were recorded in both the short axis and long axis. These images were utilized to determine left ventricular end-diastolic and end-systolic volumes (calculated as area * length * 5/6). These parameters enabled the calculation of the ejection fraction, serving as an indicator of left ventricular heart function. The investigator conducting the analysis was blinded to the genotype, gender, and age of the animals.

### 4.3. Open Field Test and Barnes Maze Experiment

The open field test was conducted following the methodology established in a previous study [48]. In this procedure, mice were gently positioned in the center quadrant of the open field arena and given 5 min to explore their surroundings. The travel trajectories of the mice were recorded using VideoMot software version 2 (TSE-Systems, Berlin, Germany). The Barnes Maze experiment was conducted following the protocol described previously [6]. The experimenters were blind to the genotypes.

### 4.4. RNA Isolation and Sequencing

RNA isolation was performed using RNA Clean and Concentrator kit (Zymo, Irvine, CA, USA) according to the manufacturer’s protocol. RNA concentration was measured with NanoDrop, and quality was evaluated using the Agilent Bioanalyzer (Agilent Technologies, Santa Clara, CA, USA). Library preparation was performed with a total of 300 ng of RNA input. For mRNA sequencing, cDNA libraries were prepared according to Illumina TruSeq, and 50 bp sequencing reads were run in HiSeq 2000. For small RNA sequencing cDNA library and sequencing have been performed according to the manufacturer’s protocol (NEBNext Small RNA library prep set for Illumina), and sequencing was performed on the HiSeq 2000 platform. For totalRNA-seq, we obtained an average sequencing depth of approx. 45 mio. reads from which 88.5% were uniquely mapped. For smallRNA-seq, we obtained an average sequencing depth of approx. 15 mio. reads from which 85.5% were uniquely mapped.

### 4.5. Bioinformatics Analysis

Sequencing data were processed using a customized in-house software pipeline. Illumina’s conversion software bcl2fastq (v2.20.2), was used for adapter trimming and converting the base calls in the per-cycle BCL files to the per-read FASTQ format from raw images. Quality control of raw sequencing data was performed by using FastQC (v0.11.5) [49]. Trimming of 3′ adapters for smallRNASeq data was performed using cutadapt (v1.11.0) [50]. The mouse genome version mm10 was used for alignment and annotation of coding and non-coding genes. MicroRNAs were annotated using miRBase [51]. For totalRNASeq, reads were aligned using the STAR aligner (v2.5.2b) [52] and read counts were generated using featureCounts (v1.5.1) [53]. For smallRNASeq, reads were aligned using the mapper.pl script from mirdeep2 (v2.0.1.2) [54] which uses bowtie (v1.1.2) [55] and read counts were generated with the quantifier.pl script from mirdeep2. All read counts were normalized according to library size to transcripts per million (TPM). Differential expression analysis was performed with the DESeq2 (v1.26.0) R (v3.6.3) package [56], here unwanted variance was removed using RUVSeq (v1.20.0) [57]. Gene Ontology (GO) enrichment analysis was performed with clusterProfiler (v4.6.0) [58]. Volcano plots were performed with the R package EnhancedVolcano (v1.12.0) [59]. Identification of interactions between miRNAs deregulated from 3 to 6 months in TG mice and genes that were deregulated in 3-month-old mice WT vs. TG were based on pairwise interactions between 221 miRNAs and 298 genes, and any other annotated targets, whose information was collected from six different databases: NPInter [60], RegNetwork [61], Rise [62], STRING Szklarczyk, 2019, TarBase [63], and TransmiR [64]. All interactions classified as weak (if available) were excluded. The Network was built using Cytoscape (v3.7.2) [65] based on selected 27 “compensatory” miRNAs and 94 genes defined as genes with reinstated expression selected from the initial interaction analysis. The list of pairwise interactors was loaded into Cytoscape to build a network. The initial network was truncated to a core network with PathLinker (v1.4.3) [66] using 1000 paths and the input list as source.

## Figures and Tables

**Figure 1 ncrna-11-00045-f001:**
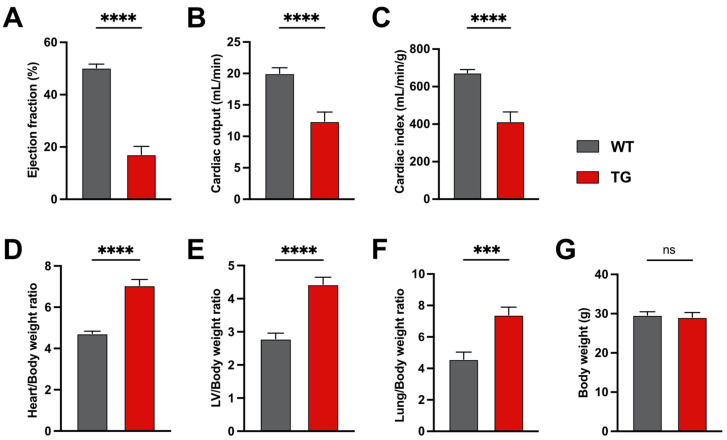
Heart Failure in 6 months CamKIIδc TG mice. (**A**) Significantly decreased ejection fraction (**B**), cardiac output, and cardiac index (**C**) in CamkIIδc TG mice (*n* = 12) compared to control mice (*n* = 16). (**D**) Weight ratios of heart to body, (**E**) left ventricle to body, and (**F**) lung to body are increased in CamkIIδc TG (*n* = 12) compared to control (*n* = 16). (**G**) No significant difference in speed, path traveled, and time spent in the middle region in the open field test between CamKIIδc TG (*n* = 11) and control (*n* = 15) mice. There were no sex-specific differences detected except for the cardiac output in the WT group (*p* = 0.04) and overall body weight (WT: *p* =< 0.0.0001; TG *p* =< 0.0001). Unpaired t-test, two-tailed; *** *p* < 0.001, **** *p* < 0.0001; Error bars indicate SEM.

**Figure 2 ncrna-11-00045-f002:**
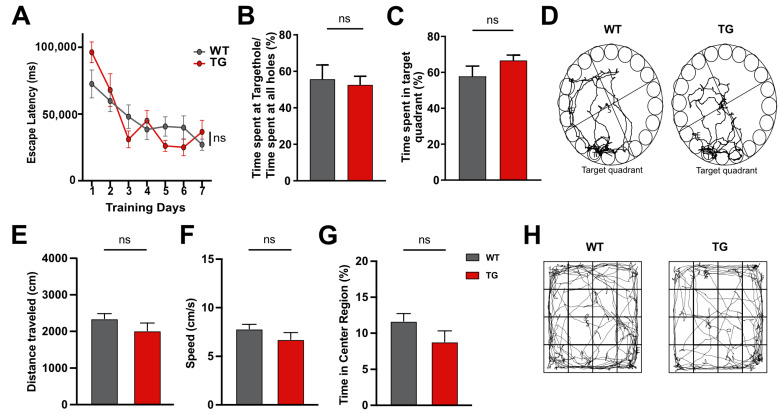
Behavioral analysis of 6-month-old CamKIIδc TG mice. (**A**) Escape latency during training sessions of the Barnes maze test is not affected in 6 old CamkIIδc TG (*n* = 11) and control mice (*n* = 15; two-way ANOVA *p* = 0.97). (**B**) Time spent at the target hole during the memory test vs. time spent at other holes is not different in 6-month-old CamkIIδc TG (*n* = 11) when compared to control mice (*n* = 15). (**C**) Time spent in the target quadrant during the memory test in fCamkIIδc TG (*n* = 11) and control mice (*n* = 15) during the memory test. (**D**) Representative images showing the path of mice during the memory test. (**E**) Distance traveled in the open field in CamkIIδc (*n* = 11) and control mice (*n* = 15). (**F**) Speed during the open field test. (**G**) Time spent in the center area of the open field is similar in CamkIIδc TG and control mice. (**H**) Representative images showing the performance during the open field test. There were no sex-specific differences detected. Unpaired t-test, two-tailed; Error bars indicate SEM.

**Figure 3 ncrna-11-00045-f003:**
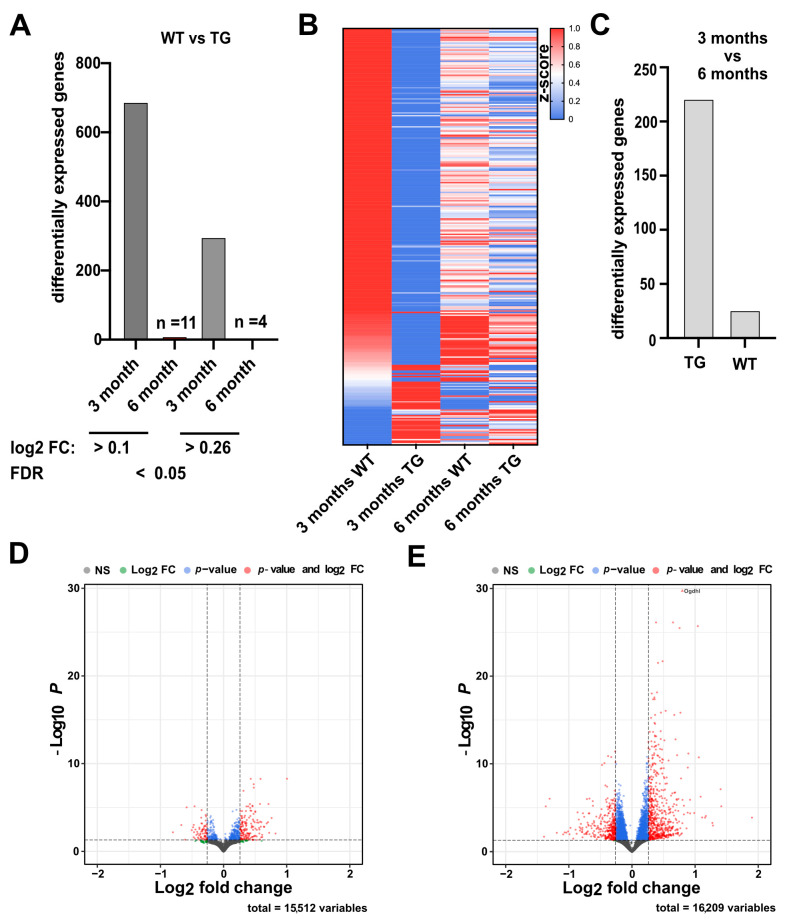
Gene expression changes in 3 and 6-month-old CamKIIδC TG mice. (**A**) Bar chart showing the number of differentially expressed genes comparing wild-type control to CamKIIδC TG mice (FDR < 0.05) at either 3 or 6 months of age. (log2FC was either < 0.1 or < 0.26). (**B**) Heatmap showing deregulated genes in 3-month-old mice in WT (*n* = 5) and CamKIIδC TG (*n* = 6) and the “rescued” gene expression in 6-month-old CamKIIδC TG mice (*n* = 11), comparable to 6-month-old WT mice (*n* = 15). (**C**) Bar chart showing the number of differentially expressed genes when 3 vs. 6 months old wild type control (WT) or 3 vs. 6 months old CamKIIδC TG mice (TG) were compared (FDR < 0.05; log2FC < 0.26). (**D**) Volcano plot showing the differentially expressed transcripts when comparing 3 vs. 6 months old wild-type control mice. (**E**) Volcano plot showing the differentially expressed transcripts when comparing 3 vs. 6 months old CamKIIδC TG mice. In (**D**,**E**), the horizontal dotted line indicates the significance threshold for the –log₁₀ *P*-value (*p* < 0.05), while the vertical dotted lines indicate the significance thresholds for the log₂ fold change (±0.26).

**Figure 4 ncrna-11-00045-f004:**
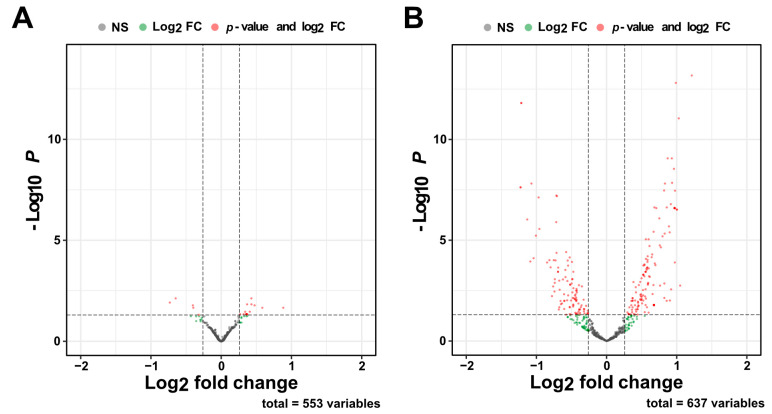
Changes in miRNA expression in wild-type control and CamKIIδC TG mice between 3 and 6 months of age. (**A**) Volcano plot showing differentially expressed miRNAs when comparing 3 vs. 6 months old wild type control mice (*n* = 5, *n* = 15, respectively). (**B**) Volcano plot showing differentially expressed miRNAs when comparing 3 vs. 6 months old CamKIIδC TG mice (*n* = 6, *n* = 16, respectively). FDR < 0.05, log2 FC < 0.26. In A and B the horizontal dotted line indicates the significance threshold for the –log₁₀ *P*-value (*p* < 0.05), while the vertical dotted lines indicate the significance thresholds for the log₂ fold change (±0.26).

**Figure 5 ncrna-11-00045-f005:**
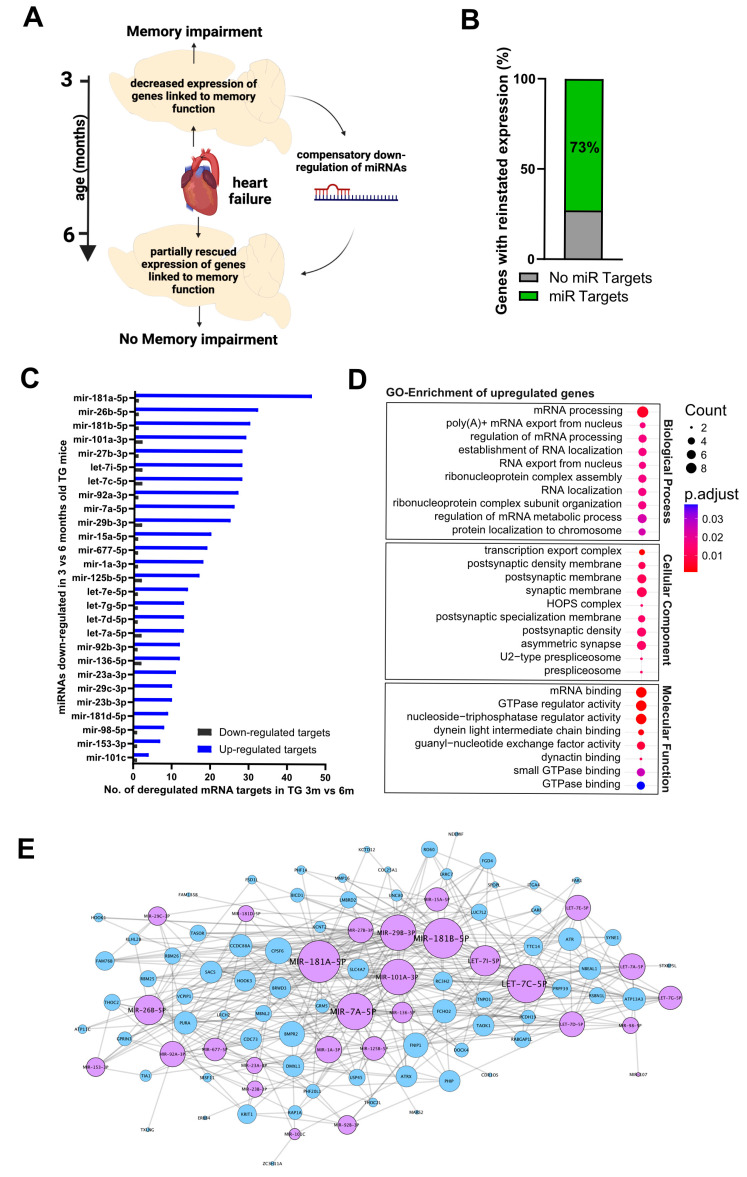
A miR RNA network that may act as a compensatory response in heart failure-mediated memory impairment. (**A**) Schematic illustration of the working hypothesis. (**B**) Bar graph showing the percentage of “rescued transcripts” targeted by the “compensatory miRNAs”. (**C**) Bar chart showing the 27 “compensatory miRNAs” (miRs with >= 5 targets) and the number of “rescued transcripts” targeted by each miRNA (blue bars). As a control (black bars), the number of mRNA transcripts upregulated in 3-month-old TG mice is shown. (**D**) Dot plot showing GO term analysis of the “rescued transcripts” upregulated from 3 to 6 months in CaMKIIδC TG mice. (**E**) Gene interaction network illustrating the 27 “compensatory miRNAs” (violet; circle size corresponds to the number of target genes regulated by each miRNA) and their relationship with the “rescued transcripts” (blue). This network accounts for 56% of the rescued transcripts.

## Data Availability

RNA sequencing data are available via the Gene Expression Omnibus (GEO) database, Accession number: GSE288827 (RNAseq) and GSE288826 (smallRNAseq). All the statistical analyses as mentioned in the main text are performed in Prism (version 9.0) or in R. All Bar Plots show Mean + SEM. Gene ontology enrichment analysis was performed using Fisher’s exact test, followed by a Benjamini–Hochberg correction.

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
