# Peer review of "Role of Compensatory miRNA Networks in Cognitive Recovery from Heart Failure"

_ncrna, 2025, doi:10.3390/ncrna11030045_

Round 1

Reviewer 1 Report

Comments and Suggestions for Authors Abstract The description of memory recovery at 6 months should clearly emphasize the significance of compensatory miRNA expression. Introduction Improve the order of the introduction. Begin with the global burden of heart failure (HF), then explore cognitive links, and finally, discuss the rationale for using a mouse model. The authors must to control typos, in Addition, put lines 75-84 in the discussion. Results Improve figure legends: Make sure legends are standalone—include sample sizes, what is being measured.
Improve figure 6.

Discussion Expand on how compensatory miRNAs compare to known plasticity mechanisms or HF recovery patterns. Add more emphasis to the model-specific nature of findings and limitations of correlational miRNA analysis.
Consider including this reference: 10.3390/cells11121882

Suggest validation (e.g., using miRNA inhibition/overexpression) to strengthen conclusions or validation with qPCR Materials & Methods Clarify animal: Detail group sizes, sex distribution (why the authors didn't a sex specific or difference analysis?), and randomization/blinding procedures explicitly. Add sequencing details: Include read depth and mapping quality metrics for RNA-seq.

Considerer adding KEGG analysis (https://doi.org/10.3390/ijms252312899)

Author Response

Reviewer 3

Reviewer 3, Comment 1

Abstract

The description of memory recovery at 6 months should clearly emphasize the significance of compensatory miRNA expression.

We now specifically refer to this issue within the abstract. It now reads:

Conclusions These findings suggest that miRNA networks contribute to the restoration of hippocampal function in HF despite continued cardiac pathology and provide an important compensatory mechanisms towards memory impairment.

Introduction

Improve the order of the introduction. Begin with the global burden of heart failure (HF), then explore cognitive links, and finally, discuss the rationale for using a mouse model. The authors must to control typos, in Addition, put lines 75-84 in the discussion.

Regarding the order of  introduction, we already follow the suggested structure. To further enhance clarity, we have introduced a column break between the paragraph discussion HF and the paragraph exploring the cognitive links. Please see page 2.

As suggested by reviewer 3 we have removed the following text from the introduction, since this issue is addressed in the discussion:

We therefore hypothesized that altered miRNA expression could drive the observed recovery of hippocampal function in 6-month-old CaMKIIδC TG mice. To test this hypothesis, we performed hippocampal RNA sequencing, and small RNA sequencing in 3- and 6-month-old CaMKIIδC TG mice and wild-type controls.

Our findings suggest that hippocampal gene-expression changes observed in 3-month-old CaMKIIδC TG mice are largely resolved by 6 months of age, paralleling the restoration of memory function. In addition, we identify a 27-miRNA signature that explains the reinstatement of the majority of genes downregulated in 3-month-old CaMKIIδC TG mice.”

Results

Improve figure legends: Make sure legends are standalone—include sample sizes, what is being measured.

Sample sizes were included in the figure legends of Figure 3 and 4, see pages 5 and 6. All changes are underlined.

Reviewer 3, Comment 2

Improve figure 6.

Since our manuscript contains only five figures, we understand that Reviewer 3 is referring to Fig. 5. For this, please see our response to Reviewer 2, Minor Comment 2. In brief, we have improved the quality and readability of Fig. 5.

Reviewer 3, Comment 3
Discussion Expand on how compensatory miRNAs compare to known plasticity mechanisms or HF recovery patterns. Add more emphasis to the model-specific nature of findings and limitations of correlational miRNA analysis.

We appreciate this comment and have now expanded the discussion to compare known plasticity mechanisms that have been linked to the recovery of memory functions in patients suffering from cardiac diseases to those observed in our study.

In the discussion, we had already included a dedicated section to address the limitations of our work. We have now expanded this section to specifically highlight the model-specific nature of our study.

Please see page 10 of the revised manuscript. Changes are underlined.

Reviewer 3, Comment 4

Consider including this reference: 10.3390/cells11121882

We have now included this reference. Please see reference 18.

Reviewer 3, Comment 5
Suggest validation (e.g., using miRNA inhibition/overexpression) to strengthen conclusions or validation with qPCR

We appreciate this comment. At present, we are not able to perform such experiments, as this would require the application of a new animal protocol. This type of research is becoming increasingly difficult in Germany, particularly in Lower Saxony, and approval would take 1–2 years. However, in the discussion, we specifically refer to relevant literature in which functional experiments on the candidate microRNAs have been conducted. For example, for one of the hub microRNAs — miR-181a-5p — there is substantial evidence suggesting that increased levels of miR-181a-5p impair memory function, while loss of miR-181a-5p function improves memory performance in rodents.

Please see page 9, last paragraph.

Materials & Methods Clarify animal: Detail group sizes, sex distribution (why the authors didn't a sex specific or difference analysis?), and randomization/blinding procedures explicitly.

Details about the group sizes are stated in the figure legends. We added this information to the figures 3 and 4 where it was not mentioned previously. Please see pages 5 and 6. Changes are underlined.

We refer to the analysis of sex specific differences now in the revised manuscript. Changes are underlined. Please see pages 3 and 4. Changes are underlined

We also state in the methods section that the experimenter was blind to the genotype. Please see page 11. Changes are underlined

Add sequencing details: Include read depth and mapping quality metrics for RNA-seq.

For totalRNA-seq we obtained an average sequencing depth of approx. 45 mio. reads from which 88.5% were uniquely mapped. For smallRNA-seq we obtained an average sequencing depth of approx. 15 mio. reads from which 85.5% were uniquely mapped. Moreover, the sequencing data is available via GEO database.

We now specifically mention this is the material & methods section. Please see page 11 of the revised manuscript. Changes are underlined.

Reviewer 3, Comment 6
Considerer adding KEGG analysis (https://doi.org/10.3390/ijms252312899)

Our analysis of the “rescued transcripts” that are targeted by our identified compensatory miRs did not reveal any significant enrichment of KEGG pathways.

Reviewer 2 Report

Comments and Suggestions for Authors

The authors of the manuscript entitled “Role of Compensatory miRNA Networks in Cognitive Recovery from Heart Failure” by Verena Gisa et al., have done a commendable job in presenting their manuscript where they investigated the molecular mechanisms linking heart failure to cognitive impairment. Additionally, the authors have focused on compensatory miRNA networks in the mouse model. The authors also showed significant memory deficits at 3 months in CaMKIIδC TG mice which reverted in 6 months. The authors identified 27 compensatory miRNAs that play a crucial role in reinstating gene expression, particularly miR-181a-5p, which is linked to improved memory function. Overall the manuscript is well written however I have some major concerns.

  1. With regards to the transcriptomic data the heatmap shows 3 months and 6 months TG but only shows the 3 month WT. I am curious to know why the 6 months Wt was excluded. Even if the transcripts were insignificant plotting them I believe is important to see the overall biology.
  2. Additionally, the volcano plots are plotted with 3months vs 6 months WT and 3months vs 6 months TG. I am curious why the authors have not compared the 3 months WT with the 3months TG. While these volcano plots show the differentical expression of genes in the same model a cross comparison analysis would be more biologically relevant.
  3. The same can be said about the figure 4 where the authors analyse the miRNA expression changes in WT and TG.
  4. With regards to the volcano plots in figure 4, The authors have highlighted the names of all the miRNAs. It would be better if the authors select a few of the significant ones as the text look too small to have them all on the plots. Additionally if one was to find the list of the miRNAs they would find in your supplementary tables. It is redundant to have it in both places.

Some of the minor comments are as follows

  1. The Figure 3E is not included in the figure legend.
  2. The fonts and the font size need to be consistent across all the images.
  3. For the Figure 4 the image looks like a cropped image some of the text such as the y axis text (P)seems to be cut off.
  4. Additional details of how many mice were used for the transcriptomic analysis and the inclusion of replicates for the heat map would be ideal. 

Author Response

Reviewer 1:

Reviewer 1 says:

The authors of the manuscript entitled “Role of Compensatory miRNA Networks in Cognitive Recovery from Heart Failure” by Verena Gisa et al., have done a commendable job in presenting their manuscript where they investigated the molecular mechanisms linking heart failure to cognitive impairment. Additionally, the authors have focused on compensatory miRNA networks in the mouse model. The authors also showed significant memory deficits at 3 months in CaMKIIδC TG mice which reverted in 6 months. The authors identified 27 compensatory miRNAs that play a crucial role in reinstating gene expression, particularly miR-181a-5p, which is linked to improved memory function. Overall the manuscript is well written however I have some major concerns.

Reviewer 1, point 1:

With regards to the transcriptomic data the heatmap shows 3 months and 6 months TG but only shows the 3 month WT. I am curious to know why the 6 months Wt was excluded. Even if the transcripts were insignificant plotting them I believe is important to see the overall biology.

We thank Reviewer 1 for the valuable suggestion. We have now included the 6-month WT condition in the heatmap. Please see Fig. 3 and its legend in the revised manuscript on page 5. Changes to the main text are underlined. 

Reviewer 1, point 2:

Additionally, the volcano plots are plotted with 3months vs 6 months WT and 3months vs 6 months TG. I am curious why the authors have not compared the 3 months WT with the 3months TG. While these volcano plots show the differentical expression of genes in the same model a cross comparison analysis would be more biologically relevant.

We agree that the comparison between 3-month WT and 3-month TG is of biological relevance. As our group focused on this comparison in a previous study (Islam et al., 2021), we did not include the corresponding volcano plot in the current study. Since our primary focus was on identifying changes between different age groups, we decided to include only volcano plots for those comparisons in the main figure. However, we have now included the comparisons between WT and TG for both age groups as a new Supplemental Figure 1, and we refer to this figure on page 4 line 149 of the main text. Changes to the main text are underlined.

Reviewer 1, point 3:

The same can be said about the figure 4 where the authors analyse the miRNA expression changes in WT and TG.

We appreciate the comment on this. Our focus was here as well the comparison between the different ages within the conditions to elucidate the potential mediators of the behavioural changes observed in the different age groups. Therefore, we decided to only include these comparisons in the main figure. However, we now included the additional comparisons of 3 months WT vs TG and 6 months Wt vs TG in Supplemental Figure 2, and we refer to this figure on page 6 lines 182-186 of the main text.

The text now reads:

To test this hypothesis, we subjected RNA isolated from the hippocampus of 3- and 6-month-old wild-type control and CamKIIδC TG mice to small RNA sequencing. We observed only minor changes in miRNA expression in 3 months-old mice comparing wild type control mice and CamKIIδC TG mice and no changes in miRNA expression in 6 months-old mice (Fig S2).

Reviewer 1, point 4:

With regards to the volcano plots in figure 4, The authors have highlighted the names of all the miRNAs. It would be better if the authors select a few of the significant ones as the text look too small to have them all on the plots. Additionally if one was to find the list of the miRNAs they would find in your supplementary tables. It is redundant to have it in both places.

We appreciated the suggestions for this figure. We optimized the figure accordingly and removed the labels of the miRNAs within the volcano plot since these can be found in the provided supplemental tables.

Reviewer 1, Minor Comments

Minor comment 1:

The Figure 3E is not included in the figure legend.

We optimized the order of the plots of Figure 3 and included the description of the heatmap (previously Figure 3E; now Figure 3B) into the figure legend. Please see the legend of Fig 3 on page 5 and page 6. All changes are underlined.

Minor comment 2:

The fonts and the font size need to be consistent across all the images.

We corrected the figures accordingly.

Minor comment 3:

For the Figure 4 the image looks like a cropped image some of the text such as the y axis text (P)seems to be cut off.

We corrected Figure 4 accordingly.

Minor comment 4:

Additional details of how many mice were used for the transcriptomic analysis and the inclusion of replicates for the heat map would be ideal.

We thank reviewer 1 for the important comment. We now added the information of number of used mice for transcriptomic analysis into the corresponding Figure legends. For RNAseq analysis please see page 5 lines and page 6. All changes are underlined.

Reviewer 3 Report

Comments and Suggestions for Authors

I would first like to congratulate the authors on quite an outstanding and novel investigation into miRNA networks in protection or recovery from heart failure-associated cognitive compromise. It was a pleasure to read such a well-written manuscript, with thorough details into the experimental approaches. Figures are generally easy to read and interpret.

I have a few minor comments.

  • Figure 4- the miRNA names are barely legible. Are you able to enlarge the text. I cannot read them still when at 120xzoom of the manuscript file.

  • Similarly, Figure 5E, is not easy to read. I might suggest making this a new figure or rearrangement of this figure so that panel 5E can be enlarged to span the whole width of the figure.

  • Y axis label on Figure 4A is cut off slightly in the manuscript version I reviewed.

  • The following sections seem to have either not been completed by the authors and/or still contain the wording from the manuscript file template:
    - Funding
  • Data Availability Statement

  • Whilst the authors make numerous outputs from the miRNA-seq analysis available to readers, I believe the raw data (FASTQ etc.) need to be made available publicly. Please include information on the data repository to which the data will be uploaded upon publication.

Author Response

Reviewer 2:

Reviewer 2 says:

I would first like to congratulate the authors on quite an outstanding and novel investigation into miRNA networks in protection or recovery from heart failure-associated cognitive compromise. It was a pleasure to read such a well-written manuscript, with thorough details into the experimental approaches. Figures are generally easy to read and interpret.

Reviewer 2, Minor Comments.

Minor comment 1:

Figure 4- the miRNA names are barely legible. Are you able to enlarge the text. I cannot read them still when at 120xzoom of the manuscript file.

We optimized Figure 4 accordingly. The labels of miRNAs in the volcano plot were removed since the list of deregulated miRNAs can also be found in the Supplemental tables.

Minor comment 2:

Similarly, Figure 5E, is not easy to read. I might suggest making this a new figure or rearrangement of this figure so that panel 5E can be enlarged to span the whole width of the figure.

We optimized Figure 5 accordingly.

Minor comment 3:

Y axis label on Figure 4A is cut off slightly in the manuscript version I reviewed.

We corrected Figure 4 accordingly.

Minor comment 3:

The following sections seem to have either not been completed by the authors and/or still contain the wording from the manuscript file template:
- Funding

  • Data Availability Statement

We thank reviewer 2 for the important comments. We removed the wording from the manuscript file template from the funding section. Please see page 12 lines 425-431.

The text now reads:

Funding: AF was supported by the DFG (Deutsche Forschungsgemeinschaft) priority program 1738, SFB1286 and GRK2824; The EU Joint Programme- Neurodegenerative Diseases (JPND) – EPI-3E; Germany’s Excellence Strategy - EXC 2067/1 390729940. FS was supported by the GoBIO project miRassay (16LW0055) by the German Federal Ministry of Science and Education. AF and KT were supported by the DZHK Innovation Cluster Brain and Heart Interfaces.  KT was  supported by a DFG Heisenberg Professorship stipend. VG was supported by the International Max Planck Research School for Genome Science.

The Data Availability section was updated including the information about accession of sequencing data on the GEO database. Please see page 12 lines 427-428.

Minor comment 4:

Whilst the authors make numerous outputs from the miRNA-seq analysis available to readers, I believe the raw data (FASTQ etc.) need to be made available publicly. Please include information on the data repository to which the data will be uploaded upon publication.

We thank the reviewer for this comment. The FASTQ files for the RNAseq and smallRNAseq are uploaded to the GEO database and will be accessible under the accession number GSE288827 (RNAseq) and GSE288826 (smallRNAseq).

Currently these can be accessed with the following tokens:

GSE288827  - efmbkukarjwrnir

GSE288826  - srexqgmwfvadzar

The Data Availability section was updated including the information about accession of sequencing data on the GEO database. Please see page 12 lines 427-428.

Round 2

Reviewer 1 Report

Comments and Suggestions for Authors

Thank you for addressing my comments and making the necessary revisions. I have reviewed the updated manuscript and I am pleased to confirm that I accept it in its current form.

Reviewer 2 Report

Comments and Suggestions for Authors

The authors have incorporated the changes.